# Laboratory Testing of Kinetic Sand as a Reference Material for Physical Modelling of Cone Penetration Test with the Possibility of Artificial Neural Network Application

**DOI:** 10.3390/ma15093285

**Published:** 2022-05-04

**Authors:** Filip Gago, Jozef Vlcek, Veronika Valaskova, Zuzana Florkova

**Affiliations:** 1Department of Geotechnics, Faculty of Civil Engineering, University of Zilina, Univerzitna 8215/1, 010 26 Zilina, Slovakia; filip.gago@uniza.sk; 2Department of Structural Mechanics and Applied Mathematics, Faculty of Civil Engineering, University of Zilina, Univerzitna 8215/1, 010 26 Zilina, Slovakia; veronika.valaskova@uniza.sk; 3Research Centre UNIZA, University of Zilina, Univerzitna 8215/1, 010 26 Zilina, Slovakia; zuzana.florkova@uniza.sk

**Keywords:** artificial neural network, cone penetration test, kinetic sand, laboratory testing

## Abstract

Cone Penetration Testing (CPT) is a quick survey in situ method through which soil parameters are not determined directly, but have to be estimated using derived relations between required soil parameter and soil resistance at the testing probe. Boundary conditions affect the reliability of the estimated soil parameters, therefore controlled laboratory conditions were applied to the intended CPT procedure analysis. Density, pycnometry, oedometer and direct shear tests of kinetic sand were performed to prove its usability as a reference testing material for further CPT laboratory analysis. The results of testing the kinetic sand are presented in this paper. Executed tests proved the kinetic sand as a reliable material in terms of the homogeneity and consistency of its physical and mechanical parameters. The material is utilizable as a substitution of cohesive sandy soils in physical modeling without the negative impact of the consistency-dependent behavior of fine-grained soils. However, some differences in parameters with respect to the natural soils should be taken into account. Neural network theory and numerical approach will be applied to the intended CPT laboratory analysis under controlled boundary conditions using kinetic sand to evaluate its potential for the determination of soil parameters.

## 1. Introduction

In situ testing represents an essential part of geotechnical surveys. Several methods can be adapted to determine the required parameters of the geological environment [1]. A recent general approach is to characterize the behavior of soil strata at original boundary conditions. These conditions involve composition and stress history which are not affected by the survey works. Laboratory sampling and testing are an important part of the survey but they can be reduced in favor of direct in situ testing methods [2]. To do so, an appropriate method should be selected to achieve reliable output. A huge part of the construction is performed in a soil-like environment [3,4]. Soil is not a homogenous isotropic material, so a certain level of knowledge about soil parameters is required for safe and economic design. Various procedures and related technical equipment were developed to determine the soil mass properties [5,6,7,8]. This paper is dedicated to the application of the static or cone penetration testing method (CPT) for soil materials.

Penetration testing, such as standard (SPT), dynamic (DPT) or static penetration testing (CPT), represents one of the quick survey in situ methods. Generally, they are based on the measurement of indirect quantities to estimate the required soil parameters. Depending on the sounding principle, the testing probe is embedded in the soil by driving or pushing. Soil resistance is expressed by the number of blows per unit advance of the probe or the corresponding resistance stress at the probe. The resistance represents the quality of the soil material. Particular characteristics, such as consistency index, deformation modulus, shear strength parameters or OCR ratio, have to be estimated according to the derived dependency between the soil resistance at the cone and the required physical and mechanical properties of soil [9,10,11,12,13,14].

Deriving the dependency requires a certain basis of data, a pair of output from in situ testing and corresponding output from, e.g., laboratory testing, to set the relation between the indirect in situ quantity and the desired soil parameter. In situ output is limited so certain soil parameters are difficult to derive. Even for the same soil type, in situ outputs from various test sites can vary because of differences in soil stratum and deposit, mineralogical composition or other irregularities. Rigorous control of data preparation for dependency estimation is necessary to exclude as many uncertainties as possible. For this reason, a large set of data is required. Dependency can be determined by the correlation between pairs of data sets but it has to be mentioned that some soil parameters cannot be determined separately with sufficient reliability. For certain soil types, the restrictions are known, and additional information about soil strata should be investigated, e.g., for clayey soils, to objectify the parameters calculated according to the dependency formula [15].

Aside from the analytical approach, artificial neural networks can be deployed to draw the relation between CPT output and real soil parameters [16,17]. As mentioned above, some soil parameters can be directly linked to the in situ results but advanced parameters, such as shear strength parameters or parameters for advanced soil material models, show much lower reliability using a direct calculation from the CPT output via derived relation formulas [13]. This led to the idea to verify the capability of the CPT method to create the data set used for the determination of such soil parameters with sufficient reliability.

Analysis of CPT method sensitivity for estimation of various soil parameters requires controlled boundary conditions. The conditions involve the testing procedure itself and the tested material. Considering the studied CPT method, the probe dimensions and pushing rate are the main control parameters. The method is suitable for testing fine-grained and sandy soils, but these soil types show the largest dispersion of parameters [16,17]. To avoid the uncertainty of such soil materials, kinetic sand was proposed as a reference testing material for future CPT sensitivity analysis and corresponding artificial neural network and numerical modeling applications [18,19,20].

## 2. Cone Penetration Testing (CPT)

Cone penetration test (CPT) was developed in the 1960s in the Netherlands and has the advantages of being a quick, nearly continuous, economical testing method. Testing of an undisturbed soil environment is one of the advantages of the method when the soil is in its original stress state and the results are not affected by the changes of the soil specimen during the sampling for laboratory tests.

The principle of the method is based on the pushing of the steel rod into the soil. A cone is attached at the end of the rod. Several types of cones can be utilized—mechanical cone (CPTm), electrical cone with measurement of pore pressures (CPTu), piezocone (SCPT), seismic piezocone (SCPTu), TDR piezocone and flat press. Additionally, the cone can be equipped with a thermometer, geophones, accelerometers, cameras or equipment for the measurement of radioisotopes, electric resistance, pH, oxygen content or fluorescence [14].

According to the cone type, the following quantities can be measured:Cone resistance (*q_c_*, *q_t_*);Shaft friction (*f_s_*);Pore pressure (*u_2_*);Velocity of shear and acoustic waves.

The parameters of the test, such as penetration depth, declination of the cone or rate of penetration, are controlled to ensure the reliability of the penetration. The testing procedure is governed by the ISO standard according to the cone type–ISO 22476-1:2012 for an electric piezocone with pore pressure measurement and ISO 22476-12:2009 for a mechanical cone [21,22].

The aforementioned quantities are used to estimate the soil properties such as consistency index or deformation modulus. Determination of the soil parameters is mainly governed by the empirical or semi-empirical relation formulas that were derived for various soil types and localities. The reliability of these formulas differs and they should be utilized carefully. Robertson suggests the usability of the CPTu testing method to determine the selected soil parameters (Table 1).

The utilization of a seismic piezocone (SCPTu) allows an increase in the estimation reliability of deformation characteristics of soil but effective strength parameters can be determined with limited reliability (Table 1).

It should be noted that the CPT probe shears the soil at higher shear strain than probes of other methods such as pressuremeter (PMT) or dilatometer (DMPT) (Figure 1). Therefore, the results are affected by the penetration rate [23].

## 3. Artificial Neural Networks (ANN)

Complex behavior of materials, such as soils, can be described by the finite number of parameters or by the algorithms for pattern recognition. Machine learning or pattern recognition techniques, such as artificial neural networks (ANN), genetic programming (GP) or fuzzy logic can be adopted for modeling of the behavior of various materials.

Numerical simulations using artificial neural networks (ANN) were developed to connect the outputs of experimental testing and numerical modeling [24,25,26,27]. This approach brings significant benefits but also implicates certain problems and disadvantages. Neural network characterization, number of neurons and hidden layers and transfer function have to be determined in advance, requiring time-consuming training procedures (Figure 2). A model based on the neural network can learn and extract the behavior of material following the experimental data. Material model based on the neural network does not require the definition of usual requirements of the elastic–plastic approach, such as plastic potential, failure function, flow rule or softening and hardening of the material.

The input layer of the network architecture is represented by the material resistance against penetration loading and another parameter of the testing procedure. Resistance consists of cone tip resistance *q_c_* and shaft friction *f_s_*. Testing parameters involve total and effective overburden pressure at given depth *σ_v_* and *σ’_v_*, respectively. The output layer of the network architecture is based on the target data such as laboratory-acquired data of the tested material. One hidden layer is sufficient for most applications. Too many neurons in the hidden layers may result in overfitting. The neural network has exceeding processing capacity and the limited amount of information in the training data set causes not all of the neurons in the hidden layers to be trained [28,29]. 

Firstly, the neural network is trained iteratively. The output of the neural network model is compared with the required targets of the training data set to calculate the error and update the weights in the hidden layer or layers. This process is performed until a minimum error is reached or the incremental improvement between iterations reaches zero [30]. The mean square error (*MSE*) can be calculated as follows:(1)$$MSE=\frac{\Sigma {\left(\hat{y}-y\right)}^{2}}{N}$$
where *ŷ* is the predicted data, *y* is the target data, and *N* is the number of samples.

In recent research, ANNs are used to classify soils or identify soil parameters from CPT or estimate the cone resistance of the CPT test [27,29,30,31].

## 4. Research Significance

Particular CPT procedures are involved in technical standards but the experiences of the authors of this paper with real cases of CPT application show a difficult approximation of regression formulas at the estimation of the soil parameters across the various test sites. In some cases, regression formulas for friction angle or cohesion derived at one test site could not be used at other test sites without some adjustment. Because of the commercial nature of the obtained data sets, these conclusions were not fully published. Considering the findings of Robertson, we cannot fully rely on the CPT potential to determine any soil parameter with sufficient reliability [14]. Aside from the tested soil stratum, CPT test procedure parameters, such as penetration rate, influence the obtained penetration resistance of the geological environment [23]. Based on these findings, we decided to evaluate the CPT procedure itself. This mainly involves the shape and the dimensions of the penetration probe and the rate of penetration.

Neural networks are used to predict various soil parameters such as soil classification, bulk density, consistency index, liquefaction potential or strength parameters based on various inputs including CPT testing output. Cone tip resistance (qc) is mainly used as a governing parameter for the estimation of the required soil characteristics [32]. The larger the data set, the better the performance of the training and validation process of the network. For example, deep neural networks and particle swarm optimization bring, together with proper preparation of data, an increase in the accuracy of the output [29]. Generally, neural networks perform well at verification but the black box nature of the results makes the evaluation of the concept hard to verify. That is also making their standalone implementation a risky process for the engineering practice [33]. The same can be stated about the estimation of soil parameters based only on the CPT output. Some parameters show good agreement between in situ and laboratory results such as bulk density or consistency index [34]. Advanced parameters of material models of soils are more test site-sensitive; thus, a detailed analysis of the CPT procedure is necessary [33]. Neural network procedures and analytical approaches can help to identify the potential and reliability of the CPT method for the estimation of soil parameters under controlled laboratory conditions.

## 5. Kinetic Sand Characterization

Sand is a natural granular material of sedimentary origin consisting of fragments. It is mainly characterized by the accumulation of grains of disturbed rocks, which have been displaced, sorted and processed to varying degrees. Minerals or rocks form the sand grains and are sufficiently resistant to chemical weathering as well as mechanical disruption during transport. Sand is an important material for the chemical, glass and construction industries, as well as a source of accumulation of some minerals and a natural reservoir of oil, natural gases and water [35].

Sand is a suitable material for physical modeling because it is not liable to change in its physical state with time. Generally, the amount of water in pores affects the physical and mechanical properties of sand but, unlike clayey soils, the sandy material can be tested in a non-saturated state without changing its properties. In situ sand is defined as soil with grain size in an interval from 0.063 to 2.0 mm according to ISO 14688-1:2002 [36]. Because of this variety, it can be used for numerous applications in physical modeling. In such a physical model, it is then possible to calibrate the real situation that we may encounter in engineering practice [37,38,39,40].

Sand is one of the soil materials that can be tested using the CPT method. With respect to the aim of the CPT method analysis, kinetic sand was proposed for laboratory testing as a reference material. Kinetic sand is a combination of fine-grained sand and an additional polymer to create a viscous–elastic material [41]. For purpose of this paper, the grained component was of main significance, but the intended utilization of kinetic sand for small scale and real scale physical modeling will also accentuate the dynamic response of the material caused to a large extent by the polymer component. The CPT method is used to measure static quantities and static soil parameters that are mainly derived from CPT’s output. Therefore, the static parameters of kinetic sand were investigated to define the boundary conditions for further CPT analysis.

Investigated kinetic sand consists of 98% sand and 2% polydimethylsiloxane which gives the kinetic sand the characteristic of viscoelasticity. The grained component of kinetic sand is represented by the siliceous sand with a nominal grain size ranging from 0.125 to 0.250 mm according to the performed sieve test (Figure 3).

Polydimethylsiloxane (PDMS) or dimethicone is a silicon-based organic polymer. It is particularly known for its rheological properties. PDMS is optically clear and, in general, inert, non-toxic, and non-flammable. The chemical formula for PDMS is CH_3_[Si(CH_3_)_2_O]*_n_*Si(CH_3_)_3_, where *n* is the number of repeating monomer [SiO(CH_3_)_2_] units (Figure 4). The density of the polymer is 965 kg·m^−3^. There is no melting or boiling point, and the material vitrifies [42,43].

Representation of chemical elements in samples of kinetic sand was investigated using an energy dispersive X-ray fluorescence spectrometer (EDXRF) which utilizes the X-ray fluorescence principle for the determination of composition and concentration of elements. The specimen is excited by the high-energy X-ray radiation. The interaction of radiation with the electron causes the ejection of an electron and its previous position is occupied by an electron from a higher energy level. Secondary radiation with a spectrum specific to the particular element is emitted (Figure 5). This radiation is analyzed and particular elements are identified step by step according to the energy of the spectral curve or position of the peaks, respectively. The concentration of elements is related to the area of peaks of the spectral curve [45].

The results of the executed spectroscopic elemental isotope analysis are plotted in Figure 6.

Silicon is the dominant element because the main part of the material consists of siliceous sand. Boiling the sand leads to a decrease in the polymer share and therefore the silicon portion (Si) is larger.

Two sets of samples were prepared for testing. Originally manufactured kinetic sand and kinetic sand were treated by boiling in demineralized water at 100 °C for 30 min to change the material properties and to lower the influence of the polymer. This resulted in the brightening of the color of the sand. The microscopic details of the sand grains of both sample types are depicted in Figure 7.

## 6. Kinetic Sand Testing

Testing of kinetic sand was aimed at the physical and mechanical parameters primarily related to the regular non-cohesive sandy soils. Both of the sample types were tested—the original kinetic sand and the boiled sample. Samples of kinetic sand without special drying were prepared for testing. Considering the output of density tests and variation of bulk density throughout the tests in oedometer and direct shear apparatus, an average value of minimal and maximal bulk density approx. 1100 ± 50 kg·m^−3^ was selected as the initial bulk density of samples.

### 6.1. Density Tests

Mechanical parameters of non-cohesive soil materials depend on the actual bulk density which can vary significantly. For technical practice, a minimal and maximal bulk density is determined as a bulk density of material treated according to the specific procedure.

A steel cylindrical container of known dimensions and weight was filled with the loose tested material without compaction (Figure 8). The minimal bulk density of the sample was calculated with a known weight and volume (Equation (2)). The same sample was then compacted with a weight lying on it on a vibration table for 8 min. The settlement of the material was measured and the new volume was calculated. The maximal bulk density was calculated in a similar way as the minimal bulk density (Equation (3)).
(2)$$\rho_{d\;min}=\frac{m_{2}{-m}_{1}}{V}$$
where *ρ_d min_* is the minimal bulk density (kg·m^−3^), *m*_2_ is the weight of the container with sample (kg), *m*_1_ is the weight of the empty container (kg), and *V* is the volume of the container (m^3^).
(3)$$\rho_{d\;\max}=\frac{m_{2}{-m}_{1}}{V_{1}}.$$
where *ρ_d max_* is the maximal bulk density (kg·m^−3^) and *V*_1_ is the changed volume of sample after compaction (m^3^).

Parameters of the sand samples are listed in Table 2. A total of two samples of each type of kinetic sand were prepared and tested.

### 6.2. Pycnometry

The pycnometer test was utilized to determine the apparent density of solid particles which is one of the inputs for the calculation of the void ratio *e*. A pycnometer of a known weight and volume was filled with the sand sample up to 1/3 of the pycnometer volume. The rest of the volume was filled with instrumental liquid, deaerated distilled water in this case, and the mixture of sand and water was mixed. The pycnometer was then warmed up to 40 °C for 40 min to deaerate the suspension. The ycnometer was fully filled with water. The apparent density of the the solid particles *ρ_s_* is calculated as follows (Equation (4)).
(4)$$\rho_{s}=\frac{\left(m_{2}{-m}_{1}\right)\cdot \rho_{l}}{V\cdot \rho_{l}{+m}_{2}{-m}_{3}}$$
where *V* is the volume of pycnometer (m^3^), *m*_1_ is the weight of empty pycnometer (kg), *m*_2_ is the weight of the pycnometer with dry sample (kg), *m*_3_ is the weight of the pycnomter with sample and instrumental liquid, and *ρ*_l_ is the density of instrumental liquid (=998 kg·m^−3^ for distilled water).

The parameters of the sand samples are shown in Table 3. A total of two samples of kinetic sand were tested. The apparent density was calculated only once because we took into account only grains of the sand and the sand was already boiled during the testing procedure regardless of the sample type. The final value of apparent density is obtained as an average of the results of particular test runs.

### 6.3. Oedometer Tests

Time-related one-dimensional settlement of the soil material is usually tested with an oedometer apparatus. The output of the test allows us to calculate the various soil parameters such as the oedometric modulus, coefficient of consolidation or swelling and compression index.

The samples of initial bulk density were prepared in the brass circular container of the oedometer apparatus (Figure 9). The initial bulk density was calculated at a known sample weight and container volume. The final bulk density was calculated considering the average sample settlement after the final loading step and following unloading.

The parameters of the sand samples are listed in Table 4. A total of three samples of original kinetic sand and three samples of boiled kinetic sand were tested.

The sample was first loaded by the weight of the piston and additional weights acting on the lever at particular loading steps (Figure 9 and Figure 10).

The same loading procedure was applied for all of the tested sample types and specimens. Loading was applied with successive steps with an increasing normal load intensity of 50, 100, 150, 200 and 250 kPa and unloading after each loading step. Only the weight of the loading piston acted on the sample during the unloading phase. A trial test was performed to determine the time interval of each loading step. The step interval was limited by the attenuation of the sample settlement. The same step interval was then applied to all of the loading steps. The average settlement of the sample was measured to calculate the final bulk density after the final loading step.

### 6.4. Direct Shear Tests

Shear strength parameters of soils represent one of the crucial inputs for the design of geotechnical structures. They can be tested using various apparatuses. Considering the non-cohesive character of the sand, a direct shear test was performed.

The sample of initial bulk density was prepared in the brass circular container of the direct shear apparatus attached to the sliding mechanism (Figure 11). The bottom side of the specimen rested on the filtration inlay. The initial bulk density was calculated at a known sample weight and container volume.

The parameters of the sand samples are displayed in Table 5. A total of four samples of original and four samples of boiled kinetic sand were tested.

Each sample was loaded by the normal consolidation stress of 50 kPa for 20 min. The sample was then loaded by horizontal sliding of the lower part of the shear box at a given rate and normal vertical stress (Figure 12).

Samples were loaded by the vertical normal stress of intensity of 50 (sample A), 100 (sample B), 150 (sample C) and 200 kPa (sample D) throughout the sliding. The rate of shear box movement was set to 0.25 mm/min with a restriction of maximum displacement to 11 mm which corresponds with the 1/10 of sample diameter. The settlement of each sample was measured to calculate the final bulk density after sliding (Figure 13).

## 7. Results

The results of the testing of the kinetic sand are presented in the following sections.

### 7.1. Density Tests

The final minimal and maximal bulk density of the sand is calculated as an average value from two concurrent determinations if the difference is less than 50 kg·m^−3^. This condition was fulfilled. In the case of the original sand, the average minimal and maximal bulk density is 918 kg·m^−3^ and 1560 kg·m^−3^, respectively. In the case of the boiled sample, the average minimal and maximal bulk density is 831 kg·m^−3^ and 1431 kg·m^−3^, respectively.

### 7.2. Pycnometry

The final apparent density of solid particles is calculated as an average value from two concurrent determinations if the difference is less than 30 kg·m^−3^. This condition was fulfilled. The apparent density of solid particles is 2465 kg·m^−3^ and this value is valid for both of the sample types of kinetic sand.

### 7.3. Oedometer Tests

Normal stress-steady settlement relation without the reversible part of the loading curve is plotted for both of the sample types in Figure 14. There is a difference between the final settlement of the sample after the final loading step in Table 4 and the final settlement of the sample in Figure 14. The final settlement in Figure 14 is only related to the loading steps without consideration of the settlement caused by the loading piston (Figure 10). Vertical normal stress introduced by the piston is approx. 4.5 kPa. Despite the small value, it causes a significant settlement of the sample that cannot be directly recorded by the sensors due to the design of the oedometer apparatus and the relative “softness” of the kinetic sand at an initial bulk density. The behavior of the kinetic sand in this stress region is negligible in terms of usual stress intensity throughout the CPT testing.

The overall trend of loading curves is similar for both of the sample types. Curves for the boiled sample show less scattering, especially in the region of higher vertical normal stresses.

#### 7.3.1. Void Ratio

The void ratio represents an important parameter describing the behavior of the material under loading. The value changes depending on the stress state and loading history. The void ratio *e* is defined as a ratio of the volume of pores *V_p_* and volume of solid particles *V_s_* and can be calculated as follows (Equations (5) and (6)).
(5)$$e=\frac{V_{p}}{V_{s}}=\frac{n}{1-n}$$
(6)$$n=1-\frac{\rho_{d}}{\rho_{s}}$$
where porosity (-), *ρ_d_* is the dry bulk density (kg·m^−3^), and *ρ_s_* is the apparent density of solid particles (kg·m^−3^). Advanced material models for numerical modeling utilize the normal stress–void ratio relation to describe the change of material characteristics under loading. This relation for the semi-logarithmic scale is plotted in Figure 15 for both of the sample types.

#### 7.3.2. Oedometric Modulus

Oedometric modulus characterizes the resistance of the material against 1-dimensional loading. Describing the deformation of the material is an important parameter for constitution material models for numerical modeling [46,47,48].

The modulus can be calculated for a given stress interval and for loading and unloading cases as follows (Equations (7) to (9)).
(7)$$\varepsilon_{i}=\frac{{\Delta h}_{i}}{h_{0}}$$
(8)$$E_{\mathrm{oed},i,i-1}=\frac{{\Delta \sigma }_{i,i-1}}{\varepsilon_{i}{-\varepsilon }_{i-1}}$$
(9)$$E_{\mathrm{oed},e,i}=\frac{{\Delta \sigma }_{i}}{\varepsilon_{i}{-\varepsilon }_{i,0}}$$
where *E_oed,i,i−1_* is the oedometric modulus for loading for stress interval between the *i-th* and *i−1-th* loading step (Mpa), *E_oed,e,i_* is the oedometric modulus for unloading for stress at the *i-th* loading step (MPa), Δ*h_i_* is the settlement of the sample at the *i-th* loading step (mm), *h_0_* is the original height of sample before first loading step (mm), *ε_i_* is the strain at loading at the *i-th* loading step (-), *ε_i−1_* is the strain at loading at the *i−1-th* loading step (-), *ε_i,0_* is the strain at unloading at the *i-th* loading step (-), Δ*σ_i,i−1_* is the stress interval at loading between the *i-th* and *i−1-th* loading steps (kPa), and Δ*σ_i_* is the stress interval at unloading at the *i-th* loading step (kPa).

Only steady settlement of the sample during the particular loading step was taken into account (Figure 15), including the unloading phase when only the loading piston acted on the surface of the sample with a vertical normal stress of 4.5 kPa. The values of the secant oedometric modulus for loading and unloading for a given stress interval are listed in Table 6. A comparison of values of the oedometric modulus of particular sand samples is also carried out for the corresponding stress intervals. The final mean was calculated for values not exceeding the 20% offset around a mean of three values. Because of a significant change in the curvature of loading curves, only the secant modulus for a particular stress interval should be evaluated.

Comparison of the oedometric modulus can be done across the sample types only for a particular stress interval. Considering the mean values, the difference in the oedometric modulus at loading between the original and the boiled kinetic sand was from 2.0% to 42.5%. The oedometric modulus of the boiled sample was lower in every stress interval. The values were more scattered at unloading phases across the particular samples of each type but the difference in means of the oedometric modulus at unloading between the original and the boiled kinetic sand was lower, from 4.5% to 32.9%. Aside from the first unloading step, the difference was in the interval from 4.5% to 12.8%.

The average value of the oedometric modulus was based on three obtained values; however, at the final loading step of the boiled sample and with the exception of one stress level in all of the unloading phases, one of the three values did not meet the 20% offset criterion. In that case, the average value is based on two input values.

### 7.4. Direct Shear Tests

Each sample type was tested at four normal stress levels. Plots of horizontal displacement of shear box Δ*L* against actual shear stress *τ* for normal stresses 50, 100, 150 and 200 kPa are shown in Figure 16. The peak shear stress was determined for each test.

Considering the Mohr–Coulomb theory, a linear regression was applied at four distinct data points in a chart of normal stress–shear relations (Figure 17). The slope of the regression curve gives the effective internal friction angle *φ’* and the intersection with the y-axis gives the effective cohesion *c’*. According to ISO 17892-10:2018 for direct shear testing, the shear stress is calculated as a ratio of horizontal shear force and initial plan area of the specimen with a note that continual change in the area of contact in the shear box is not normally taken into account [49].

## 8. Discussion

The test outputs for particular specimens of sample types show a relatively small dispersion of determined quantities with the exception of certain outputs of an oedometer test where some correction was necessary. This indicates good reliability of the results.

Boiling of the kinetic sand led to certain changes in the physical and mechanical parameters. The original sample shows an increase in minimal and maximal bulk density in comparison to the boiled sample. This difference in bulk density is the largest observed in all tests. The same relation is visible at the final bulk density after loading in the oedometer apparatus. On the other hand, the boiled sample shows a larger bulk density after testing in direct shear apparatus.

Because of the smaller nominal diameter of grains, the achieved minimal and maximal bulk densities are lower in comparison to the natural sands. Dry natural sands reach about 1600 to 1800 kg·m^−3^. The maximal bulk density of kinetic sand is closer to the bulk density of loose natural sand [50].

The apparent density shows slightly lower values in comparison to the natural clean sand, which is in an interval from 2600 to 2700 kg·m^−3^ [51].

Overall settlement of boiled sample is larger in the eodometer. The secant oedometric modulus at loading reached maximum values in a stress interval of 150–200 kPa for the original sample and in an interval of 200–250 kPa for the boiled sample. Generally, the secant oedometric modulus at loading increases with the increasing stress intensity. The unloading secant oedometric modulus can vary significantly across the sample types and particular specimens.

In the case of the original sample, there is a drop in stiffness at the final loading level of 24.1%. The final stiffness of the boiled sample is higher, but values are more scattered across the samples so values do not meet the 20% offset criterion to calculate the average from at least two values. Despite the repeated testing of the boiled sample at the stress level of 250 kPa, the values still reported a higher scattering level. Generally, boiling of the kinetic sand caused some difference in the secant oedometric modulus at loading, but, aside from the final loading step, the differences lie in the interval from 0.1 to 3.7 MPa. A higher deviation is observed at the stress levels of 200 kPa and 250 kPa in samples A and C.

The oedometric modulus for unloading shows significantly higher values in comparison with the loading phase but there is a higher dispersion across the samples. Considering the average values, the lowest value of modulus is reached at the first unloading level for both of the sample types. The deviation of the modulus for the original and boiled samples at higher stress levels lies in an interval from 1.7 to 4.2 MPa.

Generally, the stiffness of the kinetic sand is much lower in comparison to the natural sand of similar grain size graduation [52], while the void ratio is closer to minimal values [48].

Shear box displacement–shear stress relation gives more fluent propagation of curve at lower normal stress levels. A “hump” at the region of maximum shear stress is typical for soil materials with higher bulk density absences [53]. Boiling of the sand caused an increase in the cohesion with some decrease in the friction angle in comparison with the original sample. The friction angle does not deviate from the interval of peak friction angle of natural clean sand of a similar grain size and stress and compaction level [51].

Studies related to the CPT testing of sand show a stronger correlation of results for sand with a higher share of fine-grained soils. In the case of clean sand, the correlation is moderate [24]. CPT testing in sands is mainly aimed at the evaluation of the liquefaction potential, which especially poses a threat in clean sandy soils. The kinetic sand can be considered a cohesive soil according to the direct shear test output. Therefore, we assume good applicability of the kinetic sand in CPT laboratory analysis.

## 9. Conclusions and Future Research

Density, pycnometry, oedometer and direct shear tests of kinetic sand were performed to prove its usability as a reference testing material for further CPT laboratory analysis. The output of testing of the kinetic sand is presented in this paper. Executed tests proved the kinetic sand as a reliable material in terms of the homogeneity and consistency of its physical and mechanical parameters. The material is utilizable as a substitution for cohesive sandy soils in physical modeling without the negative impact of the consistency-dependent behavior of fine-grained soils. However, some differences in parameters with respect to the natural soils should be taken into account. Because of the smaller nominal diameter of grains, the bulk density and oedometric modulus show lower values in comparison with natural sands. On the other hand, its shear strength parameters are closer to regular natural sand’s at similar normal loads and compaction levels.

Neural network theory and a numerical approach will be applied to the intended CPT laboratory analysis under controlled boundary conditions using kinetic sand to evaluate its potential for the determination of soil parameters. The relation between the input and output data of the network does not have to be explicitly described by the factors with a physical meaning, but it is established according to pattern recognition. The success of the learning process is represented by the error or incremental improvement of the results. The reliability of the relation can be evaluated based on error analysis without any knowledge about the particular factors or laws governing the relation, which is a useful advantage of neural networks. These approaches help to identify the reliability of estimation of particular soil properties based on CPT output. When the test material (kinetic sand with consistent parameters) and testing conditions (penetration probe shape and dimensions and rate of penetration) are fully controlled, the potential and the sensitivity of the CPT procedure itself can be highlighted. That allows us to identify the particular soil parameters that can be determined with sufficient confidence while the derivation of these parameters can be improved at the same time with a potential exclusion or mitigation of test site influence.

## Figures and Tables

**Figure 1 materials-15-03285-f001:**
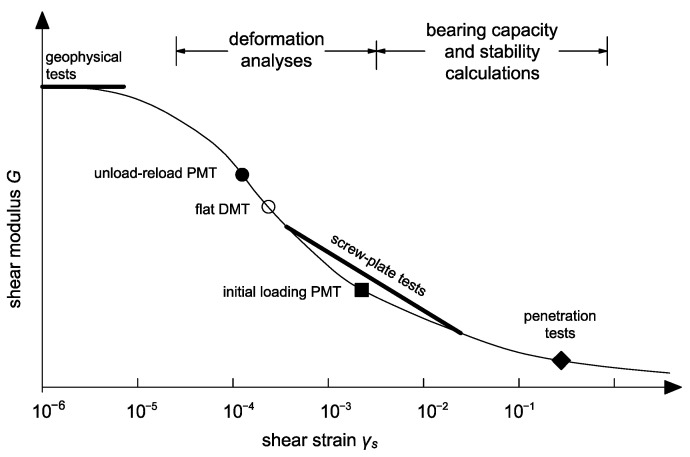
Variation of shear modulus with strain level for in situ tests [15].

**Figure 2 materials-15-03285-f002:**
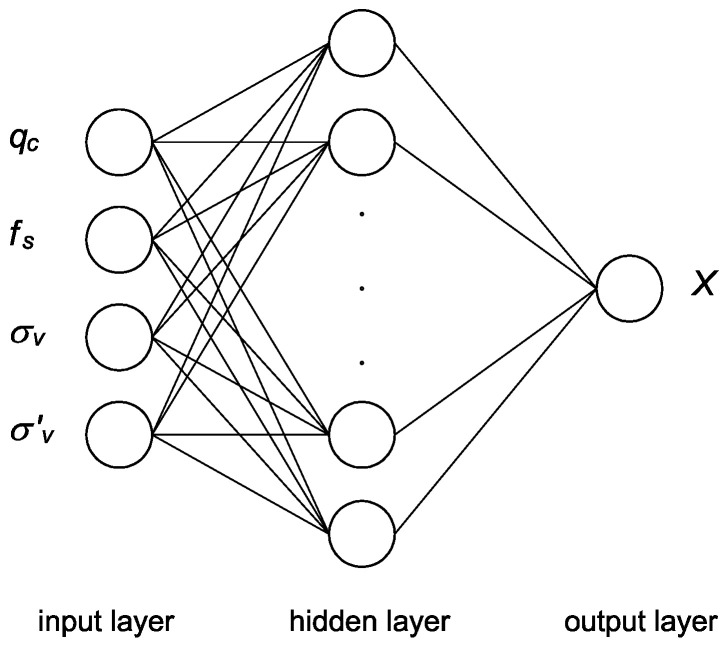
General scheme of artificial neural network architecture for machine learning for estimation of soil parameter *X* from CPT output [24].

**Figure 3 materials-15-03285-f003:**
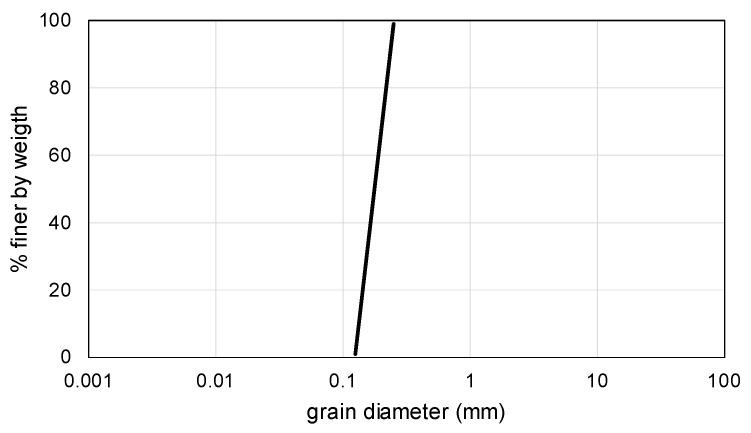
Grain size distribution curve of tested kinetic sand.

**Figure 4 materials-15-03285-f004:**
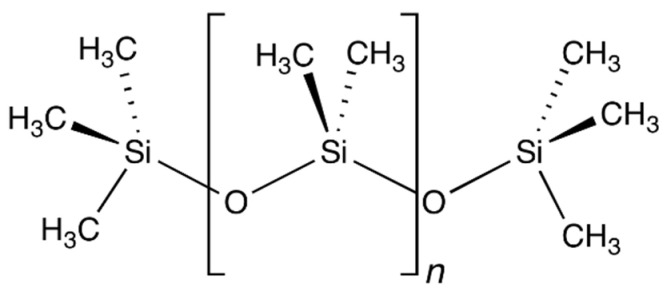
Chemical formula of polydimethylsiloxane (PDMS) [44].

**Figure 5 materials-15-03285-f005:**
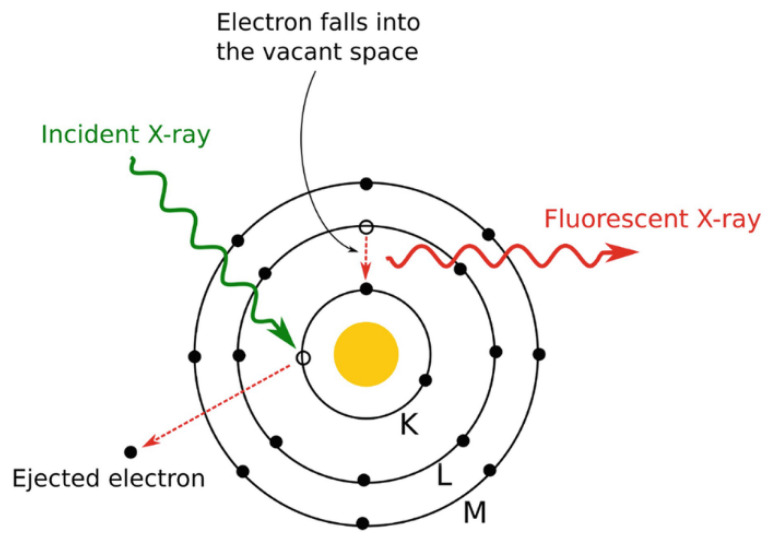
Principle of X-ray fluorescence [45].

**Figure 6 materials-15-03285-f006:**
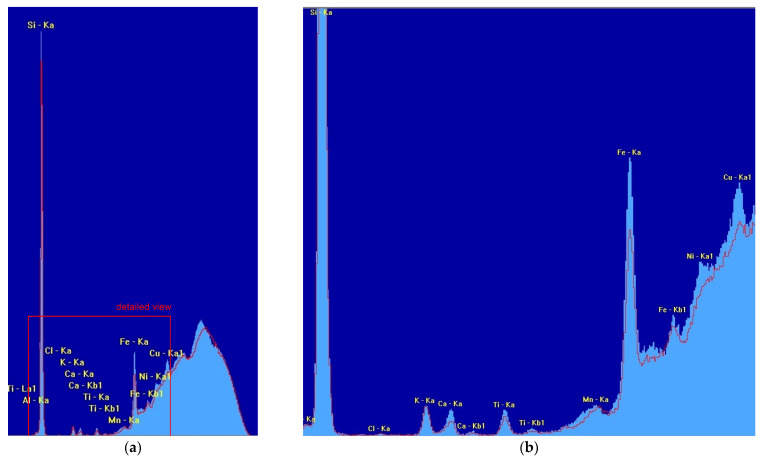
Output of spectroscopic elemental isotope analysis of kinetic sand. Red line represents the original sand. Light blue area represents the boiled sand. Si-silicon, Al-aluminum, K-potassium, Fe-iron, Ca-calcium, Ti-titanium, Ni-nickel, Mn-manganese, Cu-copper, Cl-chlorine. (**a**) Overall view; (**b**) Detailed view. Ka, Ka1, Kb1, La1—description of X-rays corresponding to the transition energy between different energy levels of atom (Ka, Ka1—transition from L level to K level, Kb1—from M to K, La1—from M to L).

**Figure 7 materials-15-03285-f007:**
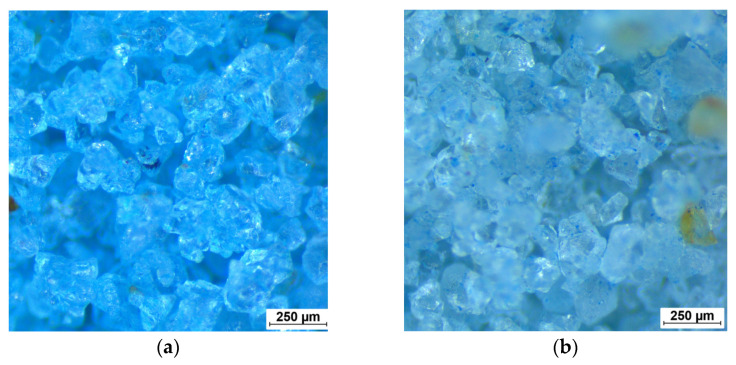
Micrographs of the kinetic sand. (**a**) Original sample; (**b**) Boiled sample. Size of sand grains observed during microscopy matches the output of sieve test.

**Figure 8 materials-15-03285-f008:**
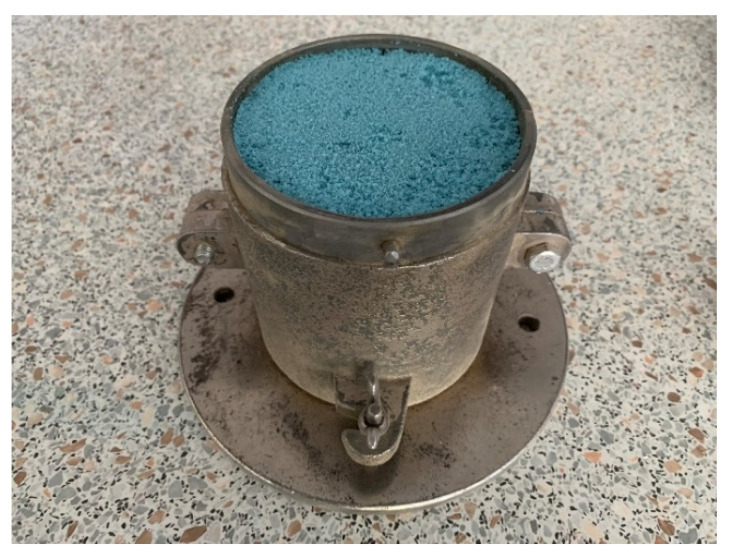
Steel cylindrical container with kinetic sand sample before compaction.

**Figure 9 materials-15-03285-f009:**
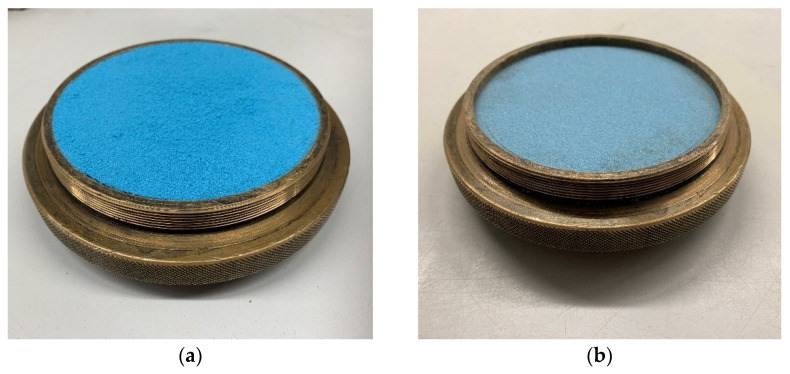
Sample of kinetic sand in the container of oedometer apparatus. (**a**) Original sample before testing; (**b**) Boiled sample after testing.

**Figure 10 materials-15-03285-f010:**
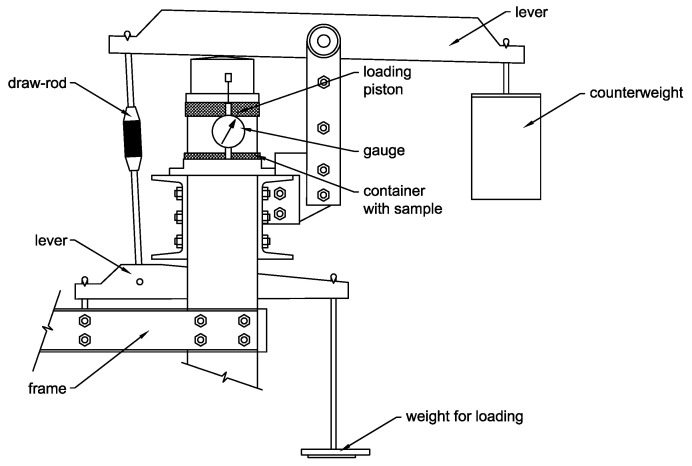
Scheme of the oedometer apparatus.

**Figure 11 materials-15-03285-f011:**
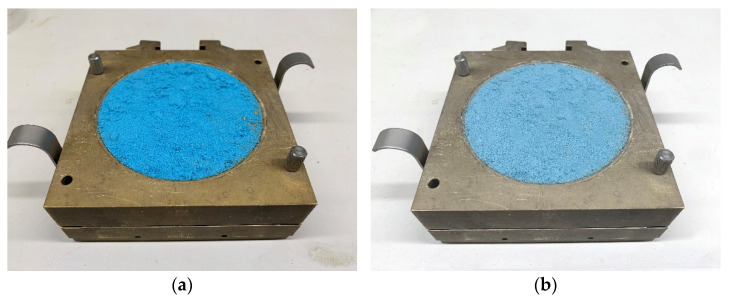
Sample of kinetic sand in the container of direct shear apparatus. (**a**) Original sample before testing; (**b**) Boiled sample before testing.

**Figure 12 materials-15-03285-f012:**
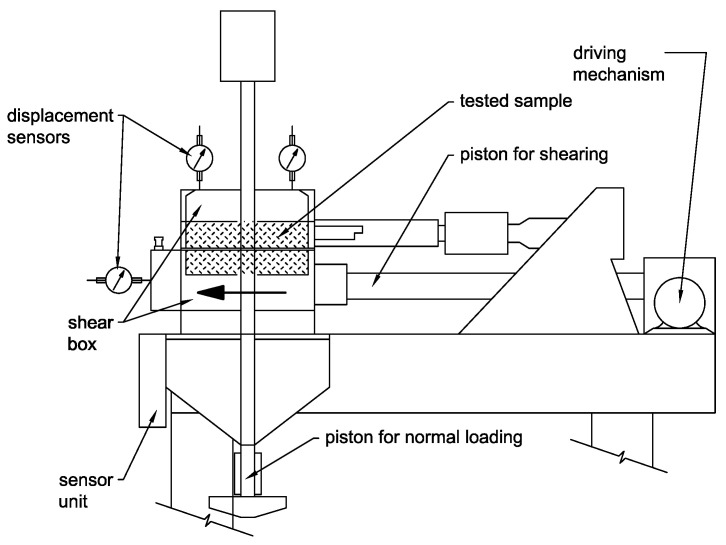
Scheme of the direct shear apparatus.

**Figure 13 materials-15-03285-f013:**
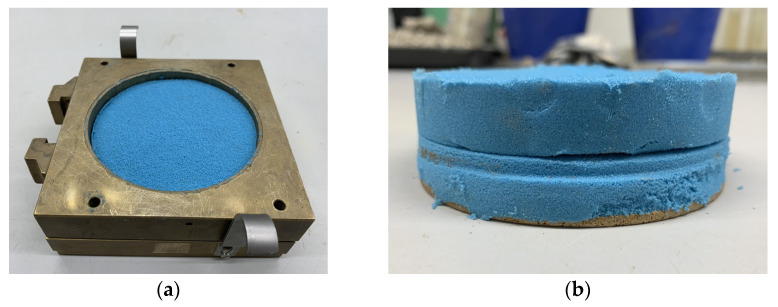
Kinetic sand after sliding. (**a**) Sample in the container; (**b**) Sample removed from the container with visible shear plane.

**Figure 14 materials-15-03285-f014:**
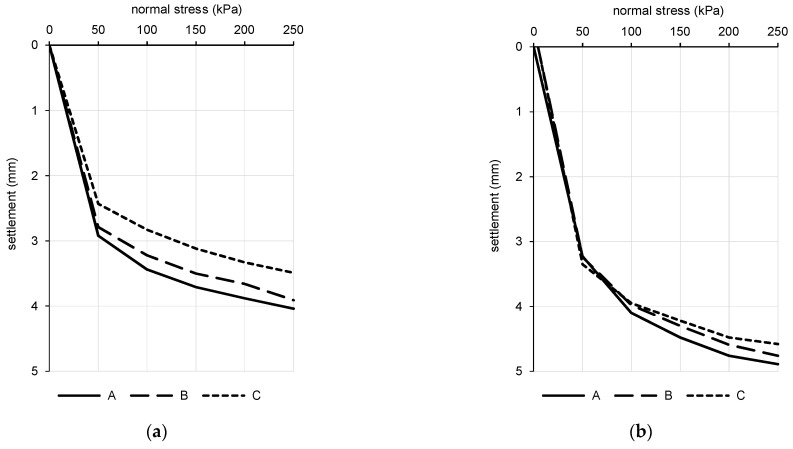
Normal stress-steady settlement relation for oedometer tests for samples A, B and C. (**a**) Original sample; (**b**) Boiled sample.

**Figure 15 materials-15-03285-f015:**
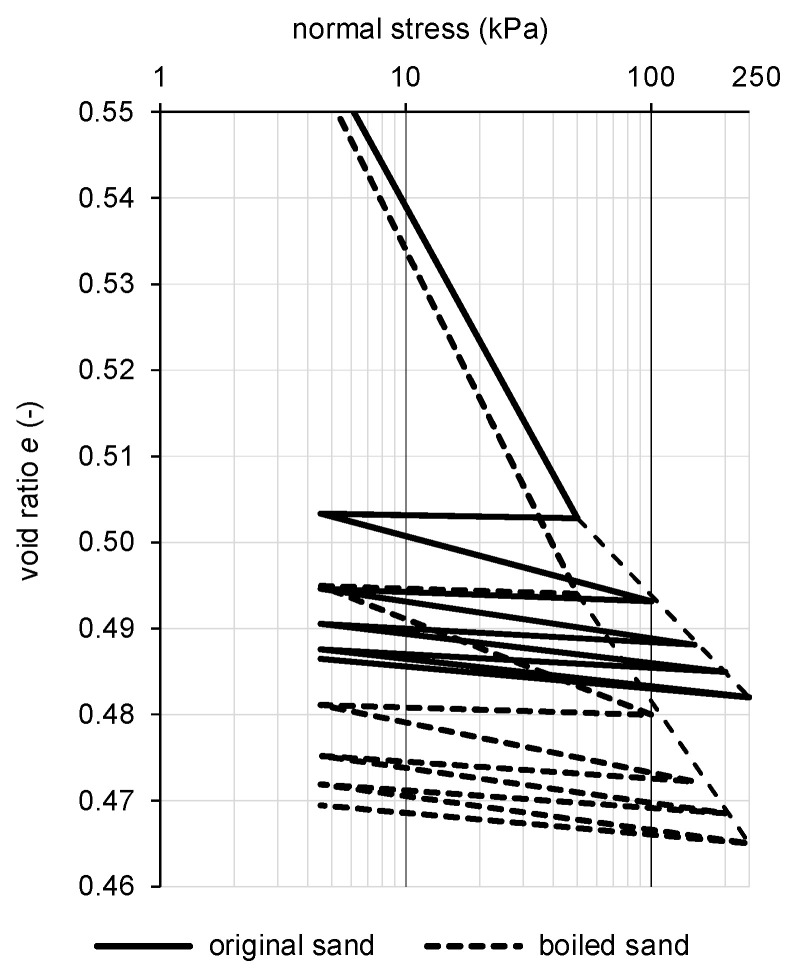
Normal stress-void ratio relation for kinetic sand (semi-logarithmic scale).

**Figure 16 materials-15-03285-f016:**
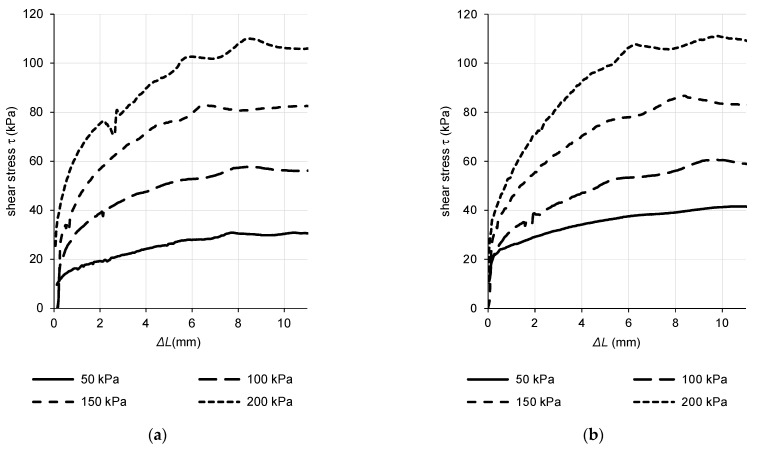
Horizontal displacement–shear stress relation. (**a**) Original sample; (**b**) Boiled sample.

**Figure 17 materials-15-03285-f017:**
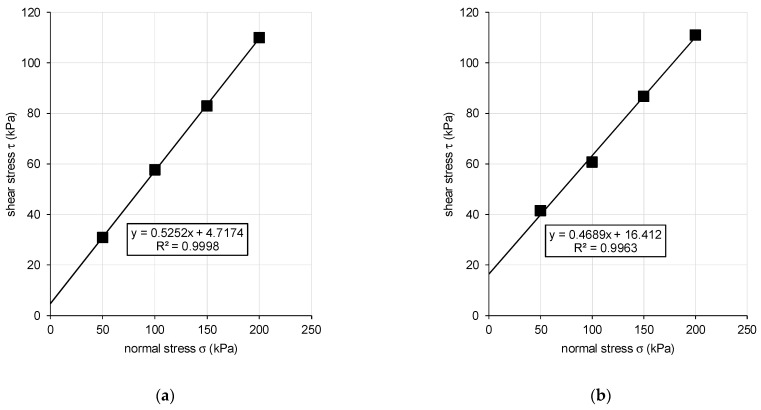
Regression curve for normal stress–shear stress relation. (**a**) Original sample; (**b**) Boiled sample. Shear strength parameters are *φ’* = 27.7° for original sand sample and *c’* = 4.7 kPa; *φ’* = 25.1° and *c’* = 16.4 kPa for boiled sand sample.

**Table 1 materials-15-03285-t001:** Usability of CPTu method to determine the soil parameters [14].

Soil Type	*I_D_*	*K* _0_	*OCR*	*s_u_*	*φ’*	*E, G **	*M*	*G_0_ **	*k*	*c_h_*
gravel, sand	2–3	5	5	-	2–3	2–3	2–3	2–3	3–4	3–4
clay	-	2	1	1–2	4	2–4	2–3	2–4	2–3	2–3

* Reliability can be increased by using the SCPTu method. *I_D_*—density index, *K_0_*—in situ earth pressure at rest, *OCR*—over consolidation ratio, *s_u_*-undrained shear strength, *φ’*—effective friction angle, *E, G*—Young’s modulus and shear modulus, *M*—1-dimensional compression, *G_0_*—small strain shear modulus, *k*—permeability, *c_h_*—coefficient of consolidation. Reliability: 1—high; 2—high to mediocre; 3—mediocre; 4—mediocre to low; 5—low.

**Table 2 materials-15-03285-t002:** Parameters of density tests.

Sample Type	Sample	Sample Diameter	Sample Height	Sample Volume	Sample Weight	Minimal Bulk Density	Average Final Settlement	Maximal Bulk Density
-	-	cm	cm	cm^3^	g	kg·m^−3^	cm	kg·m^−3^
original	A B	10.65	11.50	1024.44	875.1 869.8	941 895	4.63 4.64	1563 1557
boiled	A B	10.65	11.50	1024.44	846.9 855.6	827 835	4.91 4.73	1443 1419

**Table 3 materials-15-03285-t003:** Parameters of pycnometry tests.

Sample	Weight of Empty Pycnometer	Volume of Pycnometer	Weight of Pycnometer with Sample	Weight of Pycnometer with Sample and Water	Apparent Density
-	g	cm^3^	g	g	kg·m^−3^
A B	60.55 61.27	99.37 97.23	78.24 78.15	170.22 168.34	2470 2460

**Table 4 materials-15-03285-t004:** Parameters of oedometer tests.

Sample Type	Sample	Sample Diameter	Sample Height	Sample Volume	Sample Weight	Initial Bulk Density	Average Final Settlement	Final Bulk Density
-	-	mm	mm	cm^3^	g	kg·m^−3^	mm	kg·m^−3^
original	A B C	119.83 120.13 119.60	29.85 30.69 30.33	336.61 347.79 340.68	364 383 379	1081 1101 1112	6.63 6.39 6.09	1390 1391 1392
boiled	A B C	119.83 120.13 119.60	29.85 30.69 30.33	336.61 347.79 340.68	365 384 377	1084 1104 1107	6.00 5.93 5.55	1357 1369 1354

**Table 5 materials-15-03285-t005:** Parameters of direct shear tests.

Sample Type	Sample	Sample Diameter	Sample Height	Sample Volume	Sample Weight	Initial Bulk Density	Average Final Settlement	Final Bulk Density
-	-	mm	mm	×10^3^ mm^3^	g	kg·m^−3^	mm	kg·m^−3^
original	A B C D	99.75	35.05	273.91	311 311 312 311	1135 1135 1139 1135	7.85 8.14 7.43 7.93	1463 1479 1445 1467
boiled	A B C D	99.75	35.05	273.91	311 312 312 312	1135 1139 1139 1139	8.48 8.58 8.63 8.68	1498 1508 1511 1514

**Table 6 materials-15-03285-t006:** Oedometric modulus of kinetic sand for loading and unloading.

Sample Type	Sample	Strain *ε_i_* (-) Oedometric Modulus *E_oed,i,i−1_* (MPa)					Strain *ε**_i,0_* (-) Oedometric Modulus *E_oed,e,i_* (MPa)				
stress interval (kPa)		4.5–50	50–100	100–150	150–200	200–250	50–4.5	100–4.5	150–4.5	200–4.5	250–4.5
original	A	0.1081 0.4	0.0192 2.6	0.0100 5.0	0.0052 9.7	0.0070 7.1	0.0011 41.0	0.0030 32.2	0.0048 30.2	0.0041 48.0	0.0089 27.6
	B	0.0943 0.5	0.0154 3.2	0.0100 5.0	0.0057 8.7	0.0090 5.6	0.0022 21.1	0.0029 33.3	0.0072 20.3	0.0057 34.1	0.0061 40.3
	C	0.0876 0.5	0.0145 3.4	0.0105 4.7	0.0065 7.6	0.0069 7.2	0.0018 25.0	0.0029 32.9	0.0051 28.6	0.0058 33.6	0.0069 35.6
	mean	0.5	3.1	4.9	8.7	6.6	23.1 **	32.8	29.4 **	33.9 **	38.0 **
boiled	A	0.1135 0.4	0.0315 1.6	0.0126 4.0	0.0105 4.8	0.0046 11.0	0.0021 21.6	0.0028 34.1	0.0067 21.9	0.0077 25.4	0.0074 33.4
	B	0.1104 0.4	0.0249 2.0	0.0106 4.7	0.0109 4.6	0.0065 7.7	0.0031 14.8	0.0010 93.5	0.0044 32.9	0.0054 35.9	0.0065 37.9
	C	0.1174 0.4	0.0213 2.4	0.0086 5.8	0.0089 5.6	0.0034 14.6	0.0028 16.2	0.0024 39.8	0.0048 30.3	0.0055 35.6	0.0065 37.7
	mean	0.4	2.0	4.8	5.0	*	15.5 **	37.0 **	31.6 **	35.8 **	36.3

Original height of sample before first loading step *h_0_*: original sand: sample A = 27.02 mm; B = 27.88 mm; C = 27.52 mm, boiled sand: sample A = 28.54 mm; B = 29.36 mm; C = 29.17 mm, * values outside the 20% offset, ** average of two values.

## Data Availability

Due to the nature of this research, participants of this study did not agree for their data to be shared publicly and are only available upon reasonable request.

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
