# Peer review of "Laboratory Testing of Kinetic Sand as a Reference Material for Physical Modelling of Cone Penetration Test with the Possibility of Artificial Neural Network Application"

_materials, 2022, doi:10.3390/ma15093285_

Round 1
Reviewer 1 Report
This manuscript presented a laboratory testing of kinetic sand as a reference material for physical modelling of cone penetration test using artificial neural network. In general, the paper is well written and provide abundant information. However, there are several issues should be corrected or further clarified:
- The Abstract and conclusions should be further improved to include not only the qualitative evaluation but also the quantitative analysis.
- The state-of-art of this manuscript in this field is not balanced,some more papers published in recent years should be added.
- In conclusion remarks, some consideration about future improvement of the physical modelling of cone penetration test using artificial neural network should be reported or about the limitations of the presented study.
Reviewer 2 Report
- English is quite bad, needs double check before publishing.
- It is necessary to add details of the research's contribution to the abstract
- In the introduction, add more studies have been published literature that applied artificial neural networks to predict soil properties. For example: “Evolution of Deep Neural Network Architecture Using Particle Swarm Optimization to Improve the Performance in Determining the Friction Angle of Soil “ - https://doi.org/10.1155/2021/5570945.
- Figure 2 needs to be reproduced, in which the ANN model architecture used in the study is properly illustrated (e.g. input variables, number of layers - hidden nodes, output variables).
- Line 140, which should accurately describe all sand properties, is used as input for the ANN model.
- Finally, where does the ANN model sand's CPT prediction result in the study?
Reviewer 3 Report
Please describe the main steps that you have followed and the main outstanding outcomes in the abstract.
Please further elaborate on the novelty of your work in abstract.
Please include a brief but critical review regarding the conducted research studies in the introduction. It is recommended to add a section “research significance” and highlight the main contribution of your findings.
Please include the latest research studies related to your work preferably between 2019 and 2022.
Please include a brief summary on application of CPT test for measuring the strength of clean sands by including and referenincg the main contributions of the paper titled “Effect of Micropiles on Clean Sand Liquefaction Risk Based on CPT and SPT”
Please include statistical characteristics on the presented outcomes by the odometer tests.
Please further discuss the main parameters on characterization of the kinetic sand.
Please further explain the followed steps and the main assumptions using the artificial neural network. Accordingly please use and reference the article titled Evaluating the behaviour of centrally perforated unreinforced masonry walls: Applications of numerical analysis, machine learning, and stochastic methods
Please include statistical analyses on running the artificial neural network. For example, MSE
The presented discussion should be revised completely. Please include comparative analyses on the conducted tests such as Direct Shear Tests, Density tests and Odometer tests.
Please revise the conclusion and quantify the main outcomes in your study.
Round 2
Reviewer 2 Report
The quality of the manuscript has been improved, but the misleading title of the manuscript is building an ANN model to predict kinetic sand parameters.
Reviewer 3 Report
N/A